# Endoscopic Ultrasound-Guided Fiducial Placement for Stereotactic Body Radiation Therapy in Patients with Pancreatic Cancer

**DOI:** 10.3390/cancers15225355

**Published:** 2023-11-10

**Authors:** Irina M. Cazacu, Ben S. Singh, Rachael M. Martin-Paulpeter, Sam Beddar, Stephen Chun, Emma B. Holliday, Albert C. Koong, Prajnan Das, Eugene J. Koay, Cullen Taniguchi, Joseph M. Herman, Manoop S. Bhutani

**Affiliations:** 1Department of Oncology, Fundeni Clinical Institute, 022328 Bucharest, Romania; irina.cazacu89@gmail.com; 2Faculty of Medicine, Carol Davila University of Medicine and Pharmacy, 050474 Bucharest, Romania; 3Department of Gastroenterology, Hepatology and Nutrition, The University of Texas MD Anderson Cancer Center, Houston, TX 77030, USA; bsingh21@mdanderson.org; 4Department of Radiation Physics, The University of Texas MD Anderson Cancer Center, Houston, TX 77030, USA; rmmartin@mdanderson.org (R.M.M.-P.); abeddar@mdanderson.org (S.B.); 5Department of Radiation Oncology, The University of Texas MD Anderson Cancer Center, Houston, TX 77030, USA; sgchun@mdanderson.org (S.C.); ebholliday@mdanderson.org (E.B.H.); ackoong@mdanderson.org (A.C.K.); prajdas@mdanderson.org (P.D.); ekoay@mdanderson.org (E.J.K.); ctaniguchi@mdanderson.org (C.T.); 6Department of Radiation Oncology, Northwell Health Cancer Institute, New Hyde Park, NY 11042, USA; jherman1@northwell.edu

**Keywords:** pancreatic cancer, endoscopic ultrasound, fiducial markers, SBRT

## Abstract

**Simple Summary:**

Pancreatic cancer (PC) is known to be difficult to treat, even with standard-of-care chemotherapies. One emerging option for treating PC involves high-dose radiation therapy as a means to help control the growth and even destroy cancer cells. However, the pancreas is surrounded by many gastrointestinal organs and blood vessels; therefore, the implantation of a gold fiducial marker can help precisely target the pancreatic tumor. Our aim was to investigate whether implantation of fiducial markers through endoscopic ultrasound guidance is safe and feasible and to also document the radiation characteristics of this emerging high-dose radiation treatment option.

**Abstract:**

Accurate delivery of stereotactic body radiotherapy (SBRT) to pancreatic tumors relies on successful EUS-guided placement of fiducial markers. The aim of this study is to report the technical feasibility and safety of EUS-guided fiducial placement and to evaluate the characteristics and technical benefit of SBRT in a cohort of patients with pancreatic cancer (PC). A retrospective chart review was performed for all (*n* = 82) PC patients referred for EUS-guided fiducial placement by a single endosonographer at a tertiary cancer center. Data regarding EUS-related technical details, SBRT characteristics, adverse events, and continuous visibility of fiducials were recorded and analyzed. Most patients included in the study had either locally advanced disease (32 patients, 39%) or borderline resectable disease (29 patients, 35%). Eighty-two PC patients underwent the placement of 230 fiducial markers under EUS guidance. The technical success rate of the fiducial placement was 98%. No immediate EUS-related adverse events were reported. The average time to the simulation CT after fiducial placement was 3.1 days. Of the 216 fiducial markers used for the SBRT delivery, 202 fiducial markers were visible on both the simulation CT and the cone beam CT scan. A median dose of 40cGY was given to all the patients in five fractions. Of these, 41% of the patients reported no SBRT-related toxicities during the follow-up. Fatigue and nausea were the most reported SBRT-related toxicities, which were seen in 35% of the patients post-SBRT. Our results demonstrate that EUS-guided fiducial placement is safe and effective in target volume delineation, facilitating SBRT delivery in PC patients. Further clinical trials are needed to determine the SBRT-related survival benefits in patients with pancreatic cancer.

## 1. Introduction

Pancreatic ductal adenocarcinoma (PDAC) is considered one of the most lethal and therapeutically resistant malignancies, with a five-year survival rate of around 11% [1]. The poor prognosis is partly because, at the time of diagnosis, over 80% of patients present with locally advanced or metastatic disease. Current treatment options for pancreatic cancer include chemotherapy and radiation therapy in selected patients. Locally advanced/metastatic PDAC is particularly difficult to treat because of its poor response to chemotherapy and not being suitable for curative resection.

The use of radiation therapy (RT) in the neoadjuvant, adjuvant, or palliative setting of gastrointestinal (GI) malignancies has been continuously changing. The data suggest that RT for PDAC patients in the neoadjuvant setting (resectable/borderline resectable disease) may increase local control and the likelihood of a negative margin resection [2,3,4]. In the adjuvant setting, RT may be considered after surgery for patients with positive resection margins [5]. For locally advanced PDAC, the aim of RT is to facilitate local disease control and delay local progression [6]. Radiation therapy is reasonable for PDAC patients with metastatic disease to control symptoms such as obstruction, refractory pain, or bleeding [5].

Standard radiation therapy doses for pancreatic tumors have failed to improve survival, due to the technical and anatomic limitations of the nearby GI organs. However, delivering a high dose of radiation to pancreatic lesions has been shown to have improved overall survival and recurrence-free survival. Thus, stereotactic body radiotherapy (SBRT), which delivers high-dose radiation, has become a new therapeutic option for treating pancreatic cancer [7].

Image-guided radiotherapy (IGRT) and SBRT use advanced imaging technologies to verify and localize the target lesion during radiotherapy to decrease the treatment margins, thus decreasing toxicity to surrounding tissues. In order to safely deliver a higher dose of radiation to the pancreas while avoiding radiation toxicity to surrounding tissue, an accurate localization of the target tumor during respiration is necessary. Image-guided radiation therapy employs fiducial markers to precisely locate the target lesion during treatment and to monitor the tumor in real time, ensuring that radiation is administered with high precision. Fiducials are metallic markers, typically composed of gold or platinum, or liquid radiopaque markers that are strategically placed in close proximity to or within a target lesion. These markers serve as internal landmarks that facilitate the real-time tracking of the lesion. Most fiducial markers are gold, measuring 3 to 5 mm in length and about 1 mm in diameter, and have varying degrees of visibility in IGRT images [8]. Endoscopic ultrasound (EUS) has emerged as the preferred technique for the placement of fiducial markers. Endoscopic ultrasound has the potential to offer high-quality imaging of the internal structures located in the abdominal and mediastinal regions, thereby overcoming certain limitations associated with percutaneous insertion. Most reported EUS-guided placements of fiducials have used a 22-gauge FNA needle as the primary delivery device; however, some early studies have reported the use of 19-gauge needles, as well. Percutaneous implantation of fiducials has been reported to be safe and feasible in many studies involving the lung, liver, and pancreas; however, the reported complication rate and clinical success rate of percutaneous insertion are mixed [9,10]. Several studies have reported fiducial placement via EUS placement for gastrointestinal cancers to be more safe and feasible, with a high technical success rate, according to our recent meta-analysis that included more than 1000 patients [11]. The same analysis showed that EUS-guided fiducial placement is safe overall, with an average complication rate of 4%. Reported adverse events from EUS-guided fiducial placement included abdominal pain, mild to moderate acute pancreatitis, vomiting, minor bleeding, or elevated liver enzyme levels [11].

As the role of SBRT for palliative or curative intent for pancreatic cancer (PC) continues to emerge, EUS-guided delivery of fiducial markers will also become a more common procedure for endosonographers. Therefore, the aim of this study is to report the technical feasibility and safety of EUS-guided fiducial placement and to evaluate the technical feasibility and safety in consecutive patients with PDAC who have undergone SBRT at our institution since the launch of EUS-guided fiducial placement in 2016.

## 2. Methods

This single-center retrospective study was approved by The University of Texas MD Anderson Cancer Center’s institutional review board. A retrospective chart review was performed for all PDAC patients referred for EUS-guided fiducial placement by a single endosonographer (M.S.B.) at a tertiary cancer center (The University of Texas MD Anderson Cancer Center) between 2016 and 2022. The data were retrospectively extracted from each patient’s electronic medical records and endoscopic database, which included patient demographics, indications for fiducial placement, fiducial marker deployment system properties, EUS-related technical details, procedure- and radiation-related adverse events, and SBRT dosing/planning.

All the EUS procedures were performed using a linear-array echoendoscope (Olympus GF-UCT180, Tokyo, Japan) and the fiducials were deployed by either a commercial pre-loaded 22G fine needle aspiration (FNA) needle or were physically backloaded into standard 22G FNA needles. All the patients underwent a full EUS assessment prior to the fiducial placement in order to evaluate the tumor size and vasculatures surrounding the FNA needle trajectory. Once the tumor characteristics were identified, the FNA needle holding the fiducial marker would then be inserted into the target tumor using multiple EUS viewing planes. All the EUS procedures were given periprocedural IV antibiotics (500 mg Levaquin followed by oral antibiotics), and the patients were under deep sedation according to the institution’s standard anesthesia care guidelines. The continuous visibility of the fiducial markers was analyzed by an experienced radiation medical physicist on the simulation CT and during the final treatment fraction on the cone beam CT.

A descriptive statistical analysis was performed. All the statistical analyses were two-sided, and a *p*-value ≤ 0.05 was considered statistically significant. The distribution of continuous variables was summarized by means and standard deviations. The distribution of categorical variables was summarized using frequencies and percentages. The statistical analysis was carried out using SPSS Statistics software (https://www.ibm.com/products/spss-statistics, accessed on 16 October 2023).

## 3. Results

During the study period, a total of 82 PDAC patients underwent the placement of 230 fiducial markers under EUS guidance. The median age for the patient cohort was 69 years (range, 23–86 years), with the majority of patients being male (56%) (Table 1). Most of the patients included in the study had locally advanced disease (32 patients, 39%); 13 patients (16%) had resectable tumors, while 29 patients (35%) had borderline resectable disease. There were eight patients (10%) with metastatic disease. Most of the pancreatic tumors were located in the head (33%) or body (24%) of the pancreas.

Of the 230 fiducial markers (2.81 fiducials per patient) that were placed under EUS guidance, 166 (72%) fiducials were implanted using a backloaded technique, while 64 (28%) fiducials were implanted with a preloaded FNA needle (Table 2). Gold Anchor (Naslund Medical AB, Huddinge, Sweden) 0.28 mm × 20 mm fiducial markers were used in 64 patients (78%), and Covidien Beacon (Medtronics, Minneapolis, MN, USA) 0.43 mm × 5 mm gold fiducial markers were used in 17 patients (21%) (Figure 1. In one patient case, the LumiCoil (Boston Scientific, Boston, MA, USA) 0.46 mm × 10 mm platinum fiducial marker was used. All the fiducial placements were performed using a 22G FNA needle. Same-session EUS-guided FNA sampling was performed in 30 patients (37%).

Technical success was defined as the capability to deploy the fiducial markers in the intended location. The technical success rate of the EUS-guided fiducial placement was 98% in the current study. Technical difficulties caused by intervening blood vessels in the FNA needle pathway were noted in two (2.4%) patients. A duodenal invasion was found during pre-fiducial deployment EUS evaluation in one case, and therefore the fiducial placement was canceled. All the patients received periprocedural antibiotics, and no immediate adverse events such as perforation, pancreatitis, bleeding, or infection relating to the EUS procedure were reported.

The average time for the patients to transition from the fiducial placement to the simulation CT was 3.1 days (range, 1–15). A total of 78 (95%) patients received the SBRT at our institution following the fiducial placement (Table 3). The remaining four patients received the SBRT at their local institutions due to COVID-19 restrictions or for personal preference.

Of the 216 fiducial markers used for the CT simulation and SBRT delivery, 202 (94%) fiducial markers were clearly visible on both the CT simulation (Figure 2a) and cone beam CT scan (Figure 2b) acquired on the last day of the SBRT delivery. The other 14 (6%) fiducials were not useful for SBRT delivery, most likely due to migration or poor visibility.

The median radiation dosage for the SBRT in these patients was 40 Gy (range, 33–55) over five fractions. Thirty-two patients (41%) reported no SBRT-related toxicities during follow-up with their radiation oncologist within 2 weeks following the SBRT. The most reported SBRT-related toxicities in this patient group were fatigue and nausea, which were seen in 27 patients (35%). Seventeen patients reported mild abdominal pain, and four patients reported nausea and vomiting post-SBRT. No serious adverse events were reported relating to the SBRT.

## 4. Discussion

SBRT has emerged as a promising treatment option in pancreatic cancer care in resectable/borderline resectable, locally advanced, or palliative settings. The placement of fiducial markers directly into the pancreatic tumor and/or tumor periphery under EUS guidance is useful for targeting purposes [12,13]. While stents can aid in targeting the tumor, their reliability is lower than that of fiducials due to their tendency to shift [5].

The first case series of EUS-guided fiducial placement in patients with abdominal and mediastinal malignancies was reported by Pishvaian and colleagues [14]. Since that initial report, several studies have been published describing the safety and feasibility of EUS-guided fiducial placement in a variety of malignancies, including esophageal cancer, prostate cancer, cholangiocarcinoma, and pancreatic cancer [11]. The purpose of the current study was to assess the technical feasibility and safety of EUS-guided fiducial placement, and to evaluate the technical benefit and SBRT outcomes in a cohort of patients with PDAC treated at a tertiary cancer center.

The placement of the fiducials was determined by several factors, including patient anatomy and a multidisciplinary discussion between the endoscopist and radiation oncologist, as well as a discussion with the patient about EUS-guided placement of fiducials. Based on our current study results, the technical success rate of EUS-guided fiducial placement is very high (97%). Similarly, our previous meta-analysis of studies evaluating the technical aspects, safety, and efficacy of EUS fiducial placement in gastrointestinal malignancies, reporting on 1155 patients, revealed favorable results [11]. The technical success rate for fiducial marker placement as seen in the meta-analysis was 95% for the patients with pancreatic cancer [11]. More recent studies confirm the technical feasibility of EUS-guided fiducial placement in patients with PDAC and are in line with our results [15,16]. The main reasons for technical failure were the intervening blood vessels preventing a safe passage of the FNA needle, or the tumor was too “hard” and the fiducial marker could not exit the FNA needle. In one case, the fiducial was deployed into the duodenal wall during withdrawal but eventually migrated down the duodenum.

In the current study, a total of 230 fiducial markers were deployed; 166 (72%) fiducials were implanted using a backloaded technique, while 64 (28%) fiducials were implanted with a preloaded FNA needle. A randomized controlled clinical trial including 44 patients with PDAC showed that EUS-guided placement of preloaded fiducial markers required less time but produced similar results compared with traditional backloaded fiducials [17]. No statistically significant differences were observed between the groups in terms of technical success, the number of fiducials placed, or the incidence of adverse events [17].

No significant EUS-related adverse events were reported for the patients included in our study. The results are in line with our previous meta-analysis that yielded a low rate of adverse events (4%), indicating that EUS-guided fiducial placement is a safe procedure [11,15,18,19]. None of the studies included in the meta-analysis reported any significant adverse events, such as life-threatening bleeding or death [20,21,22,23,24,25,26]. Moreover, a recent study including 298 patients with PDAC showed that EUS-guided fiducial placement is a safe procedure, and the likelihood of infection is rare, irrespective of the administration of periprocedural antibiotics [18]. All of our patients were given 500 mg of Levaquin and were closely followed by a multi-disciplinary team of gastroenterologists and endoscopy staff following the fiducial placement; however, we did not institute a formal protocol (i.e., phone calls) for assessing immediate post-procedural complications. The referring GI radiation oncologist also evaluated the patient within a few days following fiducial placement during CT simulation and radiation planning, with no delayed EUS-related complications being reported in this patient cohort.

According to our results, 94% of the fiducial markers were clearly visible on both the CT simulation and cone beam CT scan obtained on the last day of the SBRT delivery, which was evaluated by our institution’s radiation physicist team. Only 6% of the fiducials placed under EUS guidance were not useful for SBRT delivery, most likely due to migration or poor visibility. Spontaneous migration of fiducial markers can occur; however, no migration-related adverse events were reported in our study cohort. Similarly, none of the patients in our study underwent repeat EUS for placement of additional fiducials. Spontaneous migration of fiducial markers has been reported to be related to post-procedure inflammation resolution and from the tumor-shrinking caused by successful treatment [21,24,27]. Two separate meta-analyses conducted at different times were able to report both low overall migration rates and adverse event rates relating to this endoscopic procedure [11,16]. At our institution, we aim to schedule a patient’s CT simulation 24–48 h after fiducial placement in order to capture accurate radiation planning. In our study cohort, the average time between fiducial placement and CT simulation was 3.1 days (median 2 days).

The main goals of SBRT for the study cohort patients as determined by their GI radiation oncologist were either a consolidative intent by preventing disease progression and facilitating local control for patients with locally advanced PDAC, or a curative intent for patients with resectable (not surgical candidates) or borderline resectable PDAC. The use of SBRT has been correlated with enhanced local control and progression-free survival in comparison to conventional radiation therapy. This is attributed to the ability of SBRT to administer higher radiation doses while maintaining acceptable toxicity profiles [28,29,30]. According to the most recent ASTRO consensus guideline, SBRT represents an acceptable option in patients with locally advanced pancreatic cancer, and daily image guidance with fiducial markers and volumetric imaging is recommended [31]. Of the patients who received fiducials markers, 97% completed the full course of SBRT. The remaining patients did not receive SBRT, due to disease progression or the lack of follow-up during COVID-19. No severe SBRT-related adverse event was reported; however, grade 1 symptoms of nausea and fatigue were the most commonly reported adverse events during the 10 post-radiation follow-ups.

With the advent of newer technologies that enable high-quality soft tissue imaging for IGRT, such as CT-on-rails or MRI-linac, fiducials may not be necessary [32]. In cases where cone beam CT is the only available option for radiation oncologists, it is strongly recommended that fiducials and breathing control be utilized to guarantee precise delivery of SBRT. A recent study compared the clinical outcomes of PDAC patients who received SBRT with and without fiducial markers [32]. The results suggest that the placement of fiducial markers does not negatively affect overall survival or local recurrence; moreover, the surgical outcomes were similar, irrespective of fiducial placement. The data provide reassurance that the placement of fiducials does not contribute to the progression of disease and should be employed to guarantee precise delivery of SBRT.

The limitations of this study should be noted. The results of this study are not a clear reflection of the outcomes of other hospitals, as this study was conducted retrospectively in a single tertiary cancer center. Furthermore, the results of this study should not be reflective of other anatomical sites where fiducial marker placement can be of benefit. The results may have had some minor complications omitted because there was a lag in follow-up evaluation between post-procedure recovery evaluation by the endoscopist and endoscopy staff and pre-CT simulation evaluation by the radiation oncologist. The use of three different types of fiducial markers may have skewed the visibility results, as the Medtronics and Lumicoil fiducial markers were bigger in diameter than the Gold Anchor fiducials that were used in the majority of patients. Furthermore, not standardizing the fiducial loading technique between backloading and pre-loading affected the overall length of the procedure and sedation time for all the patients, which could also affect the reporting of minor complications. It is important to note that these results should be replicated in a prospective clinical study to fully determine the objective measures of post-EUS complications, as well as radiation toxicities from SBRT targeting pancreatic lesions.

## 5. Conclusions

Accurate and precise delivery of SBRT to a pancreatic lesion relies on the successful implantation of fiducial markers via EUS guidance. EUS-guided fiducial placement is a safe and feasible procedure with a measured success rate of 97% in patients with locally advanced or borderline resectable PDAC, according to our study. Furthermore, 97% of the patients successfully completed the SBRT. Although spontaneous migration of fiducial markers can occur, the rate of fiducial migration is relatively low (5%), and no migration-related complications occurred in our study. SBRT-related toxicities during follow-up were noted in many of our patients; with fatigue and nausea being the most commonly reported complaints. No serious adverse events were reported relating to the SBRT procedure. The main obstacle to achieving effective placement of fiducial markers is the intervening vasculature. In contrast to percutaneous and intraoperative methods, EUS-guided placement offers a less invasive alternative that facilitates precise access to deep anatomic structures located in the mediastinum, abdomen, pelvis, and retroperitoneum; however, comparative studies on this matter are lacking. The placement of fiducials is an additional application for interventional EUS that has the potential to broaden the indications for SBRT by enabling access to anatomical structures that may have been previously inaccessible. Further clinical trials are needed to determine the SBRT-related survival benefits in patients specifically with pancreatic cancer.

## Figures and Tables

**Figure 1 cancers-15-05355-f001:**
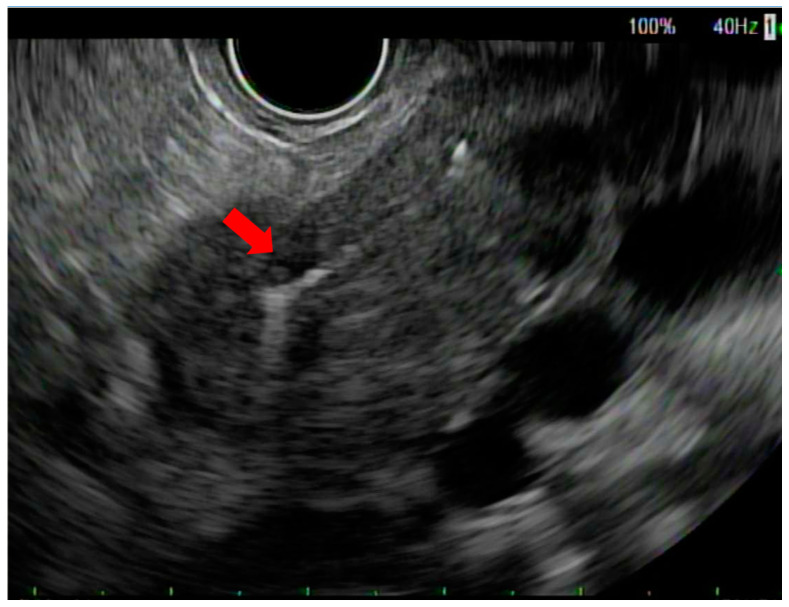
Endoscopic ultrasound view of fiducial marker in pancreatic neck tumor (red arrow).

**Figure 2 cancers-15-05355-f002:**
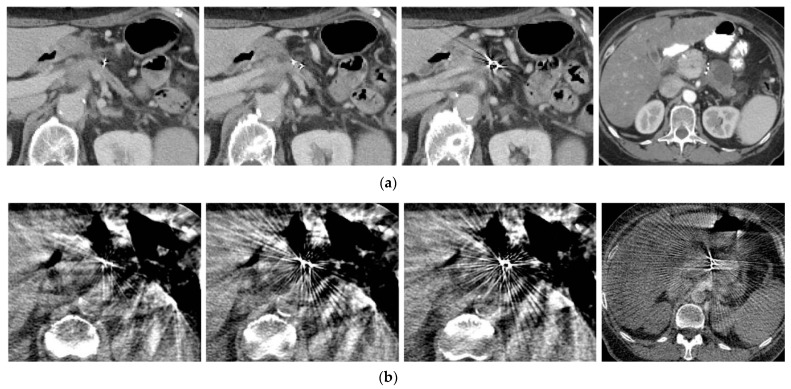
(**a**) Fiducial markers seen in pre-treatment CT simulation; (**b**) same fiducial markers as seen on cone beam CT imaging for radiation dosage planning.

**Table 1 cancers-15-05355-t001:** Patient and tumor characteristics of study population.

Gender	No. Patients (*N* = 82)
Female	36 (44%)
Male	46 (56%)
Age, Mean Year (range)	69 (23–86)
PDAC Diagnosis	
Locally Advanced	32 (39%)
Resectable	13 (16%)
Borderline Resectable	29 (35%)
Metastatic	8 (10%)
Tumor Location in Pancreas	
Head	27 (33%)
Neck	16 (20%)
Body	20 (24%)
Tail	12 (15%)
Uncinate	7 (8%)
Average Tumor Size, Largest Diameter (mm)	27

**Table 2 cancers-15-05355-t002:** EUS-guided fiducial placement procedure characteristics of study population.

Fiducial Marker Loading System Type (*N* = 230)	
Backloaded	166 (72%)
Preloaded	64 (28%)
Fiducial Marker Type Used in Patients (*N* = 82)	
Gold Anchor 20 mm Fiducial	64 (78%)
Beacon 5 mm Gold Fiducial	17 (21%)
LumiCoil Platinum Fiducial	1 (1%)
Same-Session EUS-FNA	30 (37%)
Technical Success	98%

**Table 3 cancers-15-05355-t003:** SBRT characteristics and radiation toxicity symptoms in study population.

Median Radiation Dosage over 5 Fractions	40 cGy
Visible Fiducials on Cone Beam CT (*N* = 216)	202 (94%)
Adverse Event/Symptoms Post-Radiation	
Fatigue	27 (35%)
Nausea	27 (35%)
Abdominal pain	17 (22%)
Constipation	6 (8%)
Vomiting	4 (5%)
None	32 (41%)

## Data Availability

The data presented in this study are available on request from the corresponding author.

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
