# Peer review of "Endoscopic Ultrasound-Guided Fiducial Placement for Stereotactic Body Radiation Therapy in Patients with Pancreatic Cancer"

_cancers, 2023, doi:10.3390/cancers15225355_

Round 1

Reviewer 1 Report (Previous Reviewer 1)

Comments and Suggestions for Authors

The abstract needs quantification The discussion needs modification. The conclusion may be changed to reflect the abstract content. The need for such a research may be enhanced. The limitation part needs to be modified with current trend of analysis.

Comments on the Quality of English Language

NIL

Author Response

Reviewer 2 Report (Previous Reviewer 3)

Comments and Suggestions for Authors

Indeed, the correct percentage is 2.4% based on 2 patients experiencing technical difficulties out of the 82 patients receiving fiducials- In the manuscript, it should be 2.4 not 2,4. 

The technical success rate was calculated for all patients. We did not calculate it separately for each type of fiducial marker because as you noticed the majority of the fiducials used were Gold Anchor. It was a retrospective analysis of our entire clinical experience with fiducial markers. There was a higher number of Gold Anchor fiducials because these were the type of fiducial markers available in our institution at that time.- this fact should be mentioned in the manuscript.

Author Response

This manuscript is a resubmission of an earlier submission. The following is a list of the peer review reports and author responses from that submission.

Round 1

Reviewer 1 Report

Comments and Suggestions for Authors

The abstract has to be included. The experiments and data sets are to be included in the form of tables. Review and Comparison of previous work are to be included. The methods are too much clinical with few examples. Hence, improvement of results are needed. The paper lags scientific temper with discussion and analyzing skills. The conclusion may be modified.

Comments on the Quality of English Language

NIL

Reviewer 2 Report

Comments and Suggestions for Authors

Fiducial placement for PDAC treatments is not popular, on the other hand, artifact due to metalic maerials should be avoided when weconfirm the artery invasions before and after RCT. Currently, LA-UC PDAC can be resected after CRT; therefore, evaluation after CRT is the most important factor for consideration of conversion surgery. From this background, I think artifact is the critical issue of this technique. 

Reviewer 3 Report

Comments and Suggestions for Authors

This manuscript on the safety of Endoscopic Ultrasound-Guided Fiducial Placement for Stereotactic Body Radiation Therapy in Patients with Pancreatic Cancer is well written and the conclusion is supported by the results. However, there are few concerns.

Please include an Abstract.

Please describe the statistical analysis done in detail. There is no comparison between two methods What was compared with what and which test for significance was used? This should be detailed.

Technical difficulties caused by intervening blood vessels in the FNA 119 needle pathway was noted in 2 (2%) patients.- nowhere the authors have mentioned 100 patients then how 2 patient can be 2%? Please check all %es and the number of patient or fiducials used. All the numbers in % cannot be absolute numbers, please include to two decimal point.

Table 1b- three types of fiducials were used, were the success rate was compared, or all were 100%. The authors mentioned 92% success rate. What was the reason for higher number of Gold Anchor compared to others. We cannot compare 1 with 64 to define the success rate. Please explain.

Were there any bleeding episodes? or blood in urine, stool?

Total fiducial markers placed were 230 but 216 were chosen for CT?

About 50% of the text/concepts is taken from reference number 11 which is a meta-analysis by the same author, please address this. #11 has been cited multiple times in each section.

Lines 218 onwards- text is not un same font and is bold-is there a reason for that?

Please include the limitation of the study section since this is a single center study and the patient number is only 82.
